# Self-Directed Decomposition Empowers Reasoning Potentials in Large Language Models

## Abstract

Large Language Models (LLMs) have demonstrated remarkable advancements in natural language processing and reasoning tasks, yet often struggle with logical coherence during problem-solving. This paper introduces Self-Directed Decomposition (SD), a novel prompting strategy enabling LLMs to autonomously decompose reasoning problems into manageable sub-tasks without human intervention, allowing models to determine their own approach with adaptive flexibility across diverse reasoning domains. Experiments across seven reasoning tasks reveal that this methodology particularly enhances performance on deductive, inductive, mathematical, commonsense, and scientific reasoning tasks, while showing more modest benefits for abductive and causal reasoning tasks, achieving 62.26% overall median accuracy compared to 49.64% and 46.43% for zero-shot and zero-shot Chain-of-Thought (CoT) approaches, respectively. Error and statistical analysis demonstrates that SD significantly transforms reasoning patterns by reducing wrong selection errors but increasing process mistakes for simpler variants, with only SD1 maintaining optimal balance. We discover a counterintuitive negative correlation between token consumption and accuracy ($R^2 = 0.162, p = 0.004$), challenging conventional resource-performance assumptions. Abductive reasoning demonstrates critical vulnerability to decomposition strategies, showing significant perspective errors increase ($R^2 = 0.66$). These findings explain why SD1 outperforms other variants: it balances different error types effectively while avoiding the complexity-accuracy trade-off that affects simpler decomposition strategies.

## 1  Introduction

Large Language Models (LLMs) have recently made significant progress in natural language processing (NLP), decision-making, and cognitive learning tasks (16; 18). During the LLMs evolution, zero-shot Chain-of-Thought (CoT) is a widely used prompt engineering technique to enhance LLM efficiency by instructing the model with "*Let's think step-by-step*" to break down tasks into intermediate steps (13; 30). This approach addresses complex tasks that cannot be resolved through single-step procedures into smaller, tractable step-by-step components. A significant advantage of this strategy is that it operates without necessitating pre-defined exemplars or explicit instructional guidance in the prompts. However, the main limitations of this strategy are that LLMs can still make semantic misunderstanding errors or skip intermediate steps during reasoning (24), consequently yielding erroneous conclusions and limiting their generalization capabilities.

Afterwards, based on the CoT's success, various prompting techniques have been developed to further enhance LLMs' reasoning capabilities. Including Least-to-Most prompting(36), self-consistency(26), Logical CoT (Logi-CoT)(15), Tree of Thoughts (ToT)(35), Thread of Thought (ThoT)(37), Chain of Table(29), System 2 Attention (S2A)(33), Graph of Thoughts (GoT)(2), Task Dynamic Decomposition(28). However, these approaches typically rely on predefined exemplars

or researcher-designed algorithms that constrain the model's autonomy, rather than allowing it to independently select the best analytical approaches. Researchers have also explored emotional prompting strategies that engage LLMs through emotional cues(14). Most emotional prompts require specific social contexts or additional psychological cues, except for emotional stimuli EP02, "*This is very important to my career*", which stands out as more generic.

To enhance the autonomous functioning of LLMs, this research introduces a novel prompting template, Self-Directed Decomposition (SD), which facilitates the resolution of diverse tasks without extraneous guidance. This approach enables LLMs to systematically decompose complex problems into constituent, more tractable sub-tasks based on their internal representations. To evaluate the effectiveness of SD strategies, we compare them with zero-shot, zero-shot CoT, and emotional stimuli EP02 as baseline prompts across seven reasoning datasets.

Through comprehensive experiments across multiple reasoning domains, we analyze how different SD formulations affect performance and error patterns, providing insights into the mechanisms underlying autonomous decomposition in LLMs. The remainder of this paper is structured as follows: Section 2 outlines the definitions of SD strategies. Section 3 presents experimental setups across the different datasets of reasoning tasks and the evaluation standards for the mistakes made in the processes. Section 4 shows experimental results for different prompting strategies and discusses the main reason why SD strategies perform differently with error and statistical analysis. Finally, Sections 5 and 6 summarize the main limitations and conclude this approach's implications for future research and applications and beyond.

## 2  Self-Directed Decomposition Prompting Strategies

This paper developed a series of prompting techniques called Self-Directed Decomposition (SD). These SD prompts enable LLMs to autonomously decompose complex reasoning problems into manageable sub-tasks, with varying levels of structural guidance to determine optimal decomposition complexity. SD1 includes explicit decomposition instructions using modal operators ("*should*") and structured terminology ("*sub-tasks*"), providing comprehensive guidance for the decomposition process. SD2 shortens the instruction while maintaining the core decomposition directive. SD3 adds a politeness marker ("*please*"), which may affect model behavior similar to emotional stimuli(14). SD4 adopts a minimal wording approach(9; 22; 38).

**Self-Directed Decomposition 1** *When you deal with such problems, you should do a problem decomposition like sub-tasks to analyse the problem.*

**Self-Directed Decomposition 2** *When you deal with a problem, decompose the problem and solve the task.*

**Self-Directed Decomposition 3** *Please decompose the problem and solve it.*

**Self-Directed Decomposition 4** *Decompose and solve.*

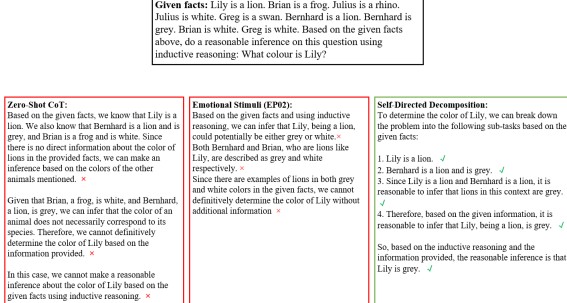

Figure 1: Self-directed decomposition solving an inductive reasoning task based on the model's understanding.

This investigation focuses on understanding how different levels of decomposition guidance affect reasoning performance and error patterns across various reasoning domains. SD1-SD4 variants test whether detailed decomposition instructions improve performance or create procedural complexity. Each variant progressively simplifies the instruction structure.

Figure 1 illustrates the decomposition approach using an inductive reasoning task as an example. This figure shows how the method systematically breaks down complex problems while maintaining logical coherence throughout the reasoning process.

# 3 Experiments

## 3.1 Datasets

These experiments aim to measure how SD strategies affect LLM performance compared to existing techniques, quantify their impact on error patterns, and identify which reasoning tasks benefit most from autonomous decomposition. The reasoning categories and datasets include:(1) **Deductive Reasoning:** A top-down approach. Starting with a general principle or assumption to conclude specific answers (8). **Datasets:** bAbI (task 15) and EntailmentBank (E-Bank) (5; 32); (2) **Inductive Reasoning:** Derives general principles based on the given observations (7; 10). **Datasets:** CLUTRR and bAbI (task 16) (21; 32); (3) **Mathematical Reasoning:** Using mathematical or logical principles to find the solution to problems (20). **Datasets:** Mathematics (Math) and SVAMP (17; 19); (4) **Scientific Reasoning:** Similar to mathematical reasoning, but using a set of collegiate-level scientific problems from calculus, chemistry and physics domains. **Dataset:** Scibench (27). (5) **Commonsense Reasoning:** Based on humans' daily knowledge and shared understanding, interpret and navigate rules and situations (1). **Dataset:** CommonsenseQA (CommonQA), PiQA and Pep-3K (4; 23; 25); (6) **Abductive Reasoning:** Infers the most likely explanation for the given observation, focusing on plausible hypotheses rather than deriving necessarily true conclusions (31). **Datasets:** $\alpha$NLI and ART (3; 7);and (7) **Causal Reasoning:** Focuses on identifying and understanding the cause-and-effect relationships among the given facts or observations (11). Datasets: E-care and Balanced-COPA (B-COPA)(6; 12).

## 3.2 Implementation

This paper used the GPT-3.5-turbo model via APIs as the backbone model for conducting the above datasets. The purpose of using this model rather than larger parameter models such as the GPT-4o series is to demonstrate that effective reasoning can be achieved through improved prompting strategies rather than simply increasing model size. This approach emphasizes that empowering a model's potential through strategic prompting may be more efficient than relying on larger architectures to overcome reasoning limitations. Each API call is dedicated to testing exactly one prompting strategy (SDs, zero-shot, zero-shot CoT, or emotional EP02) to avoid cross-contamination of information. The temperature is set to 0 throughout the experiment. From each dataset, a representative sample comprising 10 tasks was manually extracted through random selection procedures (with an expanded sample of 40 tasks from the Scibench dataset), prioritizing methodological feasibility and assessment efficiency for generalization capabilities rather than task-specific optimization paradigms. Throughout the experimental procedures, each task was presented precisely once per prompting strategy to maintain methodological consistency. Accuracy was determined based on whether the response from LLMs provided the correct answer. All statistical procedures were implemented using Python's SciPy and statsmodels libraries, with visualization support from Matplotlib and Seaborn packages. In all box plots presented in this paper, the box boundaries represent the first (**Q1**) and third (**Q3**) quartiles, the middle line shows the median (**Q2**), whiskers extend to the most extreme data points within 1.5 times the interquartile range (**IQR**) from the box boundaries, and individual points beyond the whiskers indicate outliers. All the tokens were calculated via OpenAI Tokenizer.

## 3.3 Evaluation Metrics

### 3.3.1 Error Analysis

To facilitate a more comprehensive evaluation of the performance parameters and inherent limitations of the proposed methodology, this investigation implements a systematic error analysis framework

predicated on the five-category error taxonomy established by Xu et al. (34). This error analysis is to reveal how SD strategies influence the model's reasoning process and provides insights into the underlying mechanisms through which SD affects model reasoning. The error taxonomy categorizes mistakes into five distinct types: (1) **Wrong Selection (WS)**: At the beginning of the reasoning process, LLMs choose the wrong facts or ignore the necessary facts to generate the answer. (2) **Hallucination (HA)**: LLMs incorporate information elements or factual assertions that lack empirical verification or explicitly contradict the established contexts. (3) **No Reasoning (NR)**: LLMs listed the given facts without reasoning from the facts to conclude an answer. (4) **Perspective Mistake (PPM)**: LLMs started from an incorrect or irrelevant point of view to process the answer. (5) **Process Mistake (PM)**: LLMs started from the correct perspective but made mistakes during reasoning.

### 3.3.2 Statistical Analysis

To establish statistically rigorous evaluation protocols for the accuracy and error analysis, this investigation employs multiple complementary statistical frameworks that quantify the relationships between dependent variables (error proportions, accuracy) and independent variables (prompting methods). The primary statistical metrics utilized in this investigation include: **Coefficient of Determination ($R^2$)**: The explanatory power of linear regression models is quantified using $R^2$ values, which measure the proportion of variance in error rates attributable to the independent variables. Higher $R^2$ values (approaching 1.0) indicate stronger predictive relationships, while values closer to zero indicate minimal association. **Statistical Significance ($p$-value)**: All reported correlations and regression models are subjected to hypothesis testing with a significance threshold of $\alpha = 0.05$. The $p$-values reported alongside regression analyses indicate the probability that observed relationships could occur by random chance, with lower values conferring higher confidence in the validity of the findings. **Regression Coefficients**: Linear regression equations provide quantitative measures of effect size, where the slope coefficient represents the expected unit change in error proportion per unit increase in the independent variable. These coefficients facilitate direct comparisons of effect magnitudes across different error types and reasoning categories. **Distributional Metrics**: Box plots visualize the central tendency (median), dispersion (interquartile range), and outlier characteristics of error distributions across methods and reasoning types. The positional differences in median lines and box heights quantify both the absolute performance differences and the variability in performance across experimental conditions. **ANOVA**: Analysis of variance tests determine whether observed differences in error rates across categorical variables (methods, reasoning types) reflect statistically significant patterns rather than random variation. This framework enables systematic comparisons across multiple experimental conditions while controlling for familywise error rates.

## 4 Results & Discussion

### 4.1 SD1 Achieves Superior Decomposition Performance with Optimal Error Balance

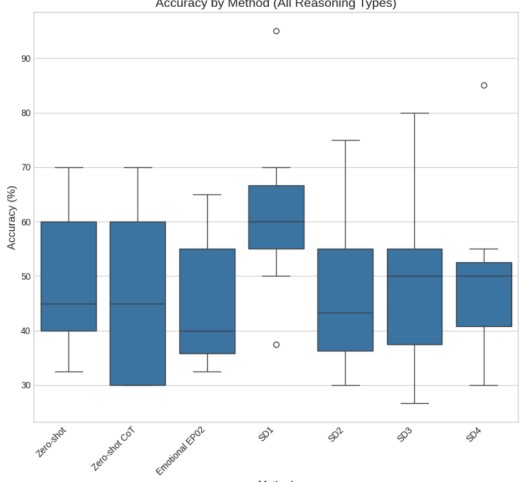

Figure 2: Overall reasoning accuracy vs. prompting methods.

As shown in Figure 2, SD1's structured decomposition approach eliminates the trade-off between accuracy and consistency that plagues other methods, achieving both superior performance (62.26%, $p < 0.05$) and tight error bounds (50%-70% IQR). This dual advantage stems from explicit decomposition instructions using modal operators ("*should*") and structured terminology ("*sub-tasks*"), which provide sufficient guidance without the excessive complexity that increases process errors

Zero-shot CoT's high variability (Q1 at 30% vs. median at 45%) exposes its fundamental limitation: the generic "*step-by-step*" instruction fails to adapt to domain-specific reasoning requirements. For instance, mathematical tasks require intermediate variable tracking and equation manipulation (e.g., solving for unknowns across multiple steps), while deductive reasoning benefits from direct premise-to-conclusion mapping without intermediate computations. This mismatch between CoT's uniform step-wise approach and domain-specific cognitive requirements explains its unstable performance across reasoning types.

The systematic performance decline from SD1 to SD4 (from 62.26% to 47.26% in median accuracy) reveals a critical threshold effect in decomposition granularity. SD2-SD4's simplified instructions lack essential structural elements—particularly modal operators and explicit task terminology—that prevent models from maintaining reasoning coherence during problem breakdown. This explains why SD2-SD4 fail: they trigger decomposition without providing the scaffolding necessary for maintaining logical consistency; instead, SD2-SD4 introduce extra information in the prompt that is meaningless for reasoning.

## 4.2  Token-Accuracy Paradox: More Computational Resources Don't Guarantee Better Performance

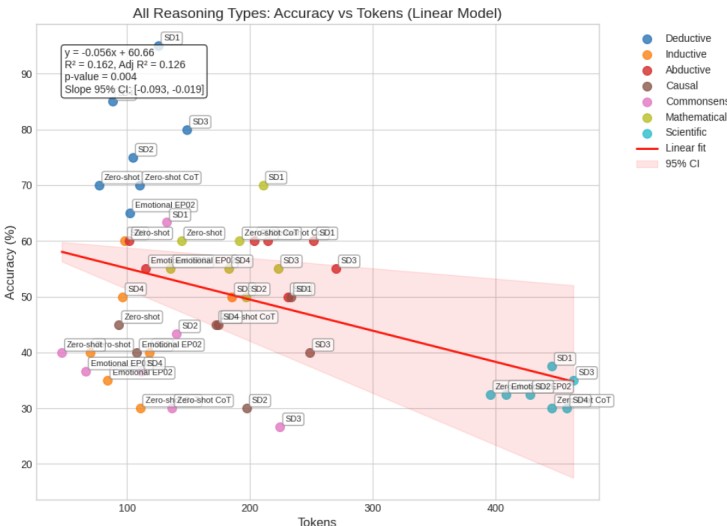

Figure 3: Overall reasoning accuracy vs. tokens.

Computational resource allocation paradoxically harms reasoning performance: higher token consumption correlates with lower accuracy ($R^2 = 0.162, p = 0.004$, slope = -0.056). This negative correlation directly contradicts the intuition that more detailed processing leads to better results—each additional token decreases accuracy by 0.056%, with statistical significance despite weak overall explanatory power (16% variance).

This paradox emerges from the mechanics of decomposition-induced error accumulation. Increased token usage typically results from excessive reasoning steps that introduce process mistakes (PM), as shown by the strong positive correlation between tokens and PM errors ($R^2 = 0.405, p < 0.001$). While longer outputs reduce wrong selection errors (WS) through better fact identification, they simultaneously create more opportunities for calculation mistakes, logical inconsistencies, and coherence failures during multi-step reasoning chains.

SD1 instances span a wide token range (100-350 tokens) while maintaining consistently low PM error rates ($< 0.3$), demonstrating unique resilience to the typical token-PM error correlation. In contrast, SD2-SD4 methods exhibit the expected positive correlation more dramatically, with PM errors escalating substantially at higher token counts. This differential response reveals that SD1's

prompt design includes inherent safeguards that prevent decomposition-induced coherence failures. Mathematical reasoning demonstrates this clearly: as shown in Figure 4, SD1 maintains PM errors below 15% despite increased complexity, while SD2-SD4 show exponential PM growth (from 30% to 35%) due to insufficient structural scaffolding. This reveals why minimal prompting fails systematically: without explicit semantic constraints, models cannot distinguish between productive decomposition (breaking coherent logical units) and counter-productive fragmentation (arbitrary text splitting). The 30% performance gap between SD1 and SD4 in mathematical tasks directly quantifies the cost of removing structural guidance mechanisms.

## 4.3 WS-PM Error Trade-off Explains Decomposition Performance Variations

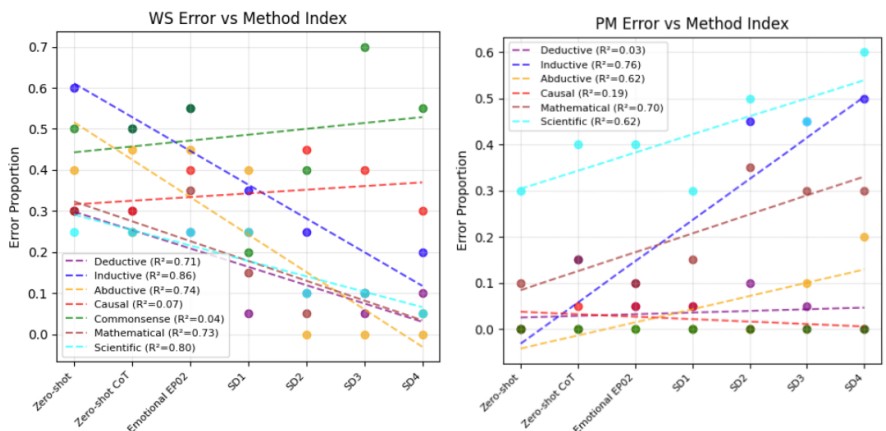

Figure 4: WS and PPM error rate vs. prompting methods.

As shown in Figure 4, SD variants systematically transform error patterns through a fundamental trade-off: decomposition reduces wrong selection (WS) errors but increases process mistakes (PM), with WS and PM errors constituting approximately 91% of all failures (WS: 61%, PM: 30%). Critically, SD variants differ in how well they manage this trade-off: SD1 maintains superior accuracy by reducing WS errors without significantly increasing PM errors, while SD2-SD4 show declining performance as PM errors overwhelm the WS improvements.

WS error reduction follows consistent patterns across most reasoning types as methods progress from zero-shot to SD variants. Deductive reasoning shows the strongest WS improvement ($R^2 = 0.71, p < 0.05$), declining from approximately 30% to 5%. Inductive, mathematical, and scientific reasoning achieve similar WS reductions ($R = 0.74 - 0.86$), while commonsense reasoning exhibits only WS errors throughout all methods, appearing solely in the WS analysis subplot of Figure 4. Notably, abductive reasoning presents an exception with increasing WS errors for SD1-SD4 variants, a phenomenon detailed in Section 4.4.

However, WS reduction comes at a cost: PM errors increase significantly with decomposition. Inductive reasoning presents the starkest trade-off ($R^2 = 0.76$), with PM errors escalating from zero values for baseline methods to 50% for SD4. Mathematical and scientific reasoning show dramatic increases ($R^2 = 0.70$ and $0.62$ respectively), with PM errors rising from approximately 10% for zero-shot to 35% for SD2 in mathematical reasoning and from 30% to 60% for SD4 in scientific reasoning.

SD1 achieves optimal balance by reducing WS errors without significantly increasing other error types, particularly PM errors. In contrast, SD2-SD4 show a clear trade-off pattern: while successfully reducing WS errors, they introduce substantial PM errors, indicating that simplification breaks the delicate balance required for effective decomposition. This trade-off creates the inverse relationship between WS and PM errors observed across reasoning types, with different domains showing varying susceptibility to process errors.

As illustrated in Figure 5, WS errors demonstrate a statistically significant negative correlation with token consumption ($R^2 = 0.131, p = 0.011$), suggesting that longer responses tend to achieve better fact selection. Scientific reasoning tasks consistently cluster in the high-token, low-error region (400+ tokens, WS < 0.3), which reflects their inherent complexity that naturally requires more elaborate

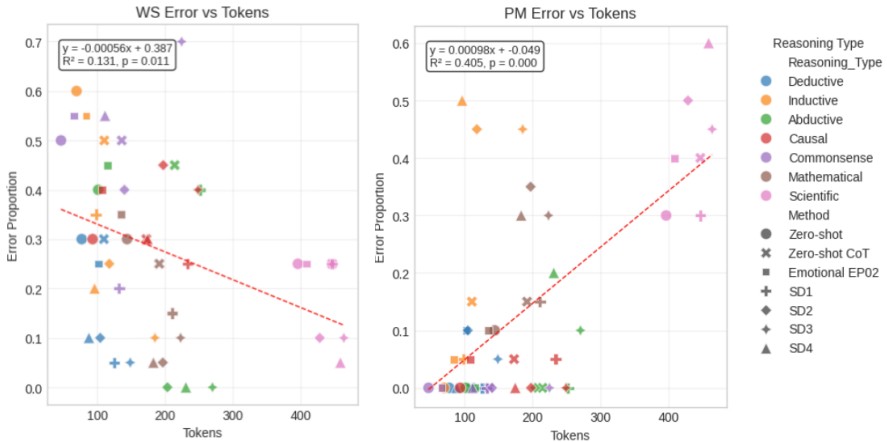

Figure 5: WS and PM error rate vs. tokens.

reasoning processes.

Conversely, PM errors exhibit a strong positive correlation with token consumption ($R^2 = 0.405, p < 0.001$, regression equation y = 0.000998x + 0.049), revealing that increased verbosity paradoxically in the response introduces more process mistakes. The stronger $R^2$ value (0.405 vs 0.131) indicates that token consumption better predicts PM errors than WS errors, demonstrating a stronger quantitative relationship between response length and process mistakes.

SD1 instances span a wide token range (100-350 tokens) while maintaining consistently low PM error rates ($< 0.2$), demonstrating unique resilience to the typical token-PM error correlation. In contrast, SD2-SD4 methods exhibit the expected positive correlation more dramatically, with PM errors escalating substantially at higher token counts. This differential response reveals that SD1's structured approach maintains coherence across varying response lengths, enabling detailed analysis without the typical increase in process mistakes.

## 4.4 Abductive Reasoning Shows Critical Vulnerability to Decomposition Strategies

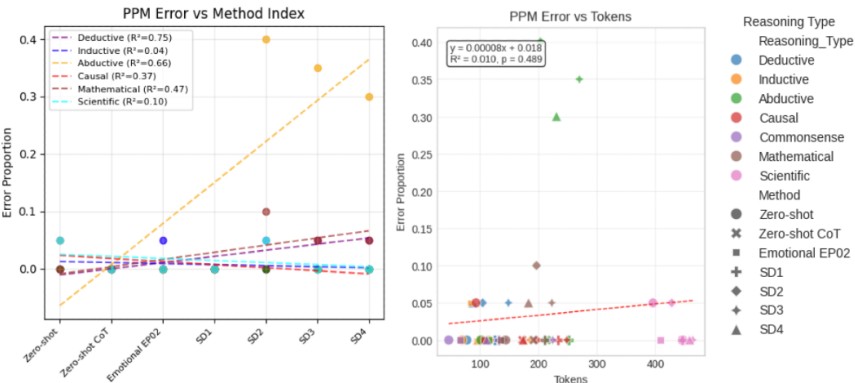

Figure 6: PPM error rate vs. prompting methods and tokens.

As demonstrated in Figure 6, Perspective Mistakes (PPM) exhibit extreme domain selectivity, with abductive reasoning showing catastrophic sensitivity to decomposition: PPM errors escalate from near-zero in zero-shot to 0.3-0.4 in SD2-SD4 ($R^2 = 0.66$). This dramatic vulnerability contrasts sharply with other reasoning types, where PPM errors remain negligible ($< 0.1$) regardless of decomposition strategy. The concentration of PPM errors in abductive reasoning reveals that inference-to-best-explanation tasks uniquely suffer when decomposition disrupts their inherent reasoning structure.

PPM errors demonstrate no correlation with token consumption ($R^2 = 0.010, p = 0.489$), fundamentally differing from WS and PM errors' token-dependent patterns. Most PPM instances cluster at near-zero error rates across all token ranges, with only abductive reasoning showing elevated values.

This independence demonstrates that PPM errors depend on decomposition structure rather than response verbosity, showing that the varying token counts produced by different SD methods have no impact on PPM rates.

Despite low absolute PPM rates in most domains, deductive ($R^2 = 0.75$) and mathematical reasoning ($R^2 = 0.47$) show strong method-dependent patterns while maintaining low error rates, indicating sensitivity to decomposition approaches without severe consequences. Abductive reasoning presents a different profile: significant method dependence ($R^2 = 0.66$) combined with elevated error rates, making it both sensitive to method changes and prone to perspective mistakes. PPM error concentration follows an inverse relationship with task structure clarity. Deductive reasoning's clear premise-conclusion structure naturally maps to sequential processing, enabling successful decomposition even with minimal guidance. Conversely, abductive reasoning requires simultaneous maintenance of competing hypotheses—a cognitive architecture that decomposition disrupts by forcing sequential evaluation of what should be parallel processes. The PPM difference between abductive and deductive reasoning under SD2-SD4 quantifies this structural incompatibility, demonstrating that effective decomposition must respect task-specific cognitive architectures rather than imposing uniform sequential processing.

The PPM analysis reveals fundamental limits of universal decomposition approaches. Abductive reasoning's unique vulnerability, combined with its independence from token consumption patterns, suggests that one-size-fits-all prompting strategies will systematically fail for certain reasoning types. Future decomposition frameworks should incorporate domain-specific guards against PPM errors, particularly for inference tasks requiring simultaneous consideration of multiple explanatory hypotheses.

This selective vulnerability challenges assumptions about decomposition effectiveness across reasoning domains. The concentration of PPM errors in abductive reasoning, despite its structural simplicity compared to mathematical or scientific tasks, suggests that task difficulty and decomposition compatibility are orthogonal dimensions. This finding has profound implications for the development of general-purpose reasoning systems, indicating that effective prompting requires domain-aware architectural considerations beyond simple instruction complexity.

# 5   Limitations

While this investigation demonstrates SD1's potential for enhancing autonomous problem-solving capabilities, with notable improvements in deductive reasoning (95% accuracy on deductive reasoning) and up to 25% gains over baseline approaches, several important limitations warrant consideration. These constraints contextualize our findings and highlight opportunities for future research.

The error analysis reveals domain-specific limitations, particularly in PPM error patterns for abductive reasoning and the systematic trade-off between WS and PM errors. While SD1 maintains an optimal balance, the substantial performance decline in SD2-SD4 variants suggests that decomposition strategies are sensitive to structural prompt modifications. Moreover, the negative correlation between token consumption and overall accuracy ($R^2 = 0.162$, $p = 0.004$) indicates that computational efficiency does not uniformly translate to performance gains across reasoning types.

Our analysis reveals a fundamental tension where increased token consumption reduces WS errors but increases PM errors. This relationship suggests inherent limitations in current decomposition frameworks, where the very mechanisms designed to improve fact selection create conditions favoring process mistakes. The high variance in performance across reasoning types indicates that domain-specific optimizations may be necessary rather than universal decomposition strategies.

The performance ceiling observed in complex domains like scientific reasoning (37.5% maximum accuracy) suggests that SD strategies remain bounded by the underlying LLMs' knowledge and architectural limitations. The inconsistent patterns in abductive reasoning across datasets (e.g., SD2's 100% accuracy on ART versus poor performance of 20% accuracy on $\alpha$NLI) further highlight the limitations of current decomposition approaches for certain reasoning structures.

These limitations suggest several promising avenues: investigating domain-adaptive decomposition strategies that balance WS-PM error trade-offs, developing hybrid approaches for reasoning types sensitive to PPM errors, and exploring the integration of SD methods with other prompting techniques. Additionally, robustness testing under varied real-world conditions—including input noise, domain variations, and assumption violations—represents a critical area for future investigation.

## 6  Conclusion

The Self-Directed Decomposition (SD) methodology enhances LLMs' autonomous reasoning capabilities by instructing models to decompose complex problems independently, without relying on human intervention or exemplar-based prompting. Unlike previous approaches that constrained models to predefined problem-solving trajectories, SD enables models to autonomously develop problem-solving strategies based on their understanding of tasks.

Experimental results across seven reasoning domains reveal that SD1 achieves superior performance by maintaining an optimal balance between error types, while subsequent variants (SD2-SD4) exhibit deteriorating performance despite reduced structural complexity. The comprehensive error analysis uncovers a fundamental trade-off in decomposition reasoning: while SD strategies systematically reduce wrong selection (WS) errors through improved fact selection, they simultaneously introduce process mistakes (PM) that increase with computational resource allocation. SD1's exceptional performance stems from its resilience to this token-error paradox, maintaining consistent error control across varying token consumption levels.

The investigation further reveals domain-specific error patterns, particularly the concentration of perspective mistakes (PPM) in abductive reasoning, highlighting the limitations of universal decomposition prompting strategies. The negative correlation between overall token consumption and accuracy ($R^2 = 0.162, p = 0.004$) demonstrates that increased computational resources do not uniformly improve performance, challenging the intuitive assumption that more detailed processing leads to better results—a relationship our findings directly contradict.

These findings contribute to a nuanced understanding of decomposition-based prompting, revealing both its transformative potential and inherent limitations. The work demonstrates that effective prompt engineering requires careful calibration between methodological sophistication and error control, with different reasoning types demanding specialized optimization strategies. Future research should focus on developing adaptive frameworks that can dynamically adjust decomposition complexity based on domain characteristics, error risk profiles, and the fundamental token-error trade-offs identified in this investigation.

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

# NeurIPS Paper Checklist

1. **Claims**

   Question: Do the main claims made in the abstract and introduction accurately reflect the paper's contributions and scope?

   Answer: [Yes]

   Justification: The paper's abstract and introduction clearly articulate the main claims and contributions. The abstract (lines 1-21) introduces Self-Directed Decomposition as a novel prompting strategy that enables LLMs to autonomously decompose reasoning problems without human intervention, and accurately states the performance improvement (62.26% overall median accuracy compared to 49.64% and 46.43% for baseline approaches). The introduction (lines 22-57) further elaborates on the limitations of existing prompting techniques, explains how the SD approach differs by allowing models to determine their own problem-solving approach with adaptive flexibility, and outlines the paper's methodology for evaluating across seven diverse reasoning tasks. Both sections present realistic claims about the approach's capabilities and limitations, noting challenges in abductive reasoning tasks, providing readers with an accurate representation of the research scope and contributions.

   Guidelines:

   - The answer NA means that the abstract and introduction do not include the claims made in the paper.
   - The abstract and/or introduction should clearly state the claims made, including the contributions made in the paper and important assumptions and limitations. A No or NA answer to this question will not be perceived well by the reviewers.
   - The claims made should match theoretical and experimental results, and reflect how much the results can be expected to generalize to other settings.
   - It is fine to include aspirational goals as motivation as long as it is clear that these goals are not attained by the paper.

2. **Limitations**

   Question: Does the paper discuss the limitations of the work performed by the authors?

   Answer: [Yes]

   Justification: The paper includes a comprehensive limitations section (Section 5, lines 285-310) that discusses several key constraints of the proposed methodology. It acknowledges performance ceilings in complex scenarios like scientific reasoning (37.5% maximum accuracy), addresses inconsistent patterns in specific reasoning types (particularly abductive reasoning showing sensitivity to decomposition strategies), and discusses the fundamental trade-off between wrong selection and process mistake errors. The paper also notes that SD strategies remain bounded by the underlying LLMs' knowledge and architectural limitations, and highlights the importance of domain-specific optimizations rather than universal decomposition strategies. This section provides important context for understanding the boundaries of the Self-Directed Decomposition approach and guides future research directions.

   Guidelines:

   - The answer NA means that the paper has no limitation while the answer No means that the paper has limitations, but those are not discussed in the paper.
   - The authors are encouraged to create a separate "Limitations" section in their paper.
   - The paper should point out any strong assumptions and how robust the results are to violations of these assumptions (e.g., independence assumptions, noiseless settings, model well-specification, asymptotic approximations only holding locally). The authors should reflect on how these assumptions might be violated in practice and what the implications would be.
   - The authors should reflect on the scope of the claims made, e.g., if the approach was only tested on a few datasets or with a few runs. In general, empirical results often depend on implicit assumptions, which should be articulated.
   - The authors should reflect on the factors that influence the performance of the approach. For example, a facial recognition algorithm may perform poorly when image resolution

is low or images are taken in low lighting. Or a speech-to-text system might not be used reliably to provide closed captions for online lectures because it fails to handle technical jargon.

- The authors should discuss the computational efficiency of the proposed algorithms and how they scale with dataset size.

- If applicable, the authors should discuss possible limitations of their approach to address problems of privacy and fairness.

- While the authors might fear that complete honesty about limitations might be used by reviewers as grounds for rejection, a worse outcome might be that reviewers discover limitations that aren't acknowledged in the paper. The authors should use their best judgment and recognize that individual actions in favor of transparency play an important role in developing norms that preserve the integrity of the community. Reviewers will be specifically instructed to not penalize honesty concerning limitations.

3. **Theory assumptions and proofs**

   Question: For each theoretical result, does the paper provide the full set of assumptions and a complete (and correct) proof?

   Answer: [NA]

   Justification: The paper focuses on empirical validation of the Self-Directed Decomposition methodology through extensive experiments across seven reasoning domains rather than developing mathematical theorems or formal proofs. The work presents experimental findings and qualitative analysis of performance patterns, but does not claim theoretical guarantees that would require formal mathematical proofs. Therefore, this question is not applicable to the current research

   Guidelines:

   - The answer NA means that the paper does not include theoretical results.
   - All the theorems, formulas, and proofs in the paper should be numbered and cross-referenced.
   - All assumptions should be clearly stated or referenced in the statement of any theorems.
   - The proofs can either appear in the main paper or the supplemental material, but if they appear in the supplemental material, the authors are encouraged to provide a short proof sketch to provide intuition.
   - Inversely, any informal proof provided in the core of the paper should be complemented by formal proofs provided in appendix or supplemental material.
   - Theorems and Lemmas that the proof relies upon should be properly referenced.

4. **Experimental result reproducibility**

   Question: Does the paper fully disclose all the information needed to reproduce the main experimental results of the paper to the extent that it affects the main claims and/or conclusions of the paper (regardless of whether the code and data are provided or not)?

   Answer: [Yes]

   Justification: The paper provides comprehensive information for reproducing the main experimental results. Section 3.2 (Implementation, lines 102-120) details the model used (GPT-3.5-turbo), API configuration (temperature set to 0), experimental procedures, and evaluation methodology. The paper clearly describes the sampling approach for each dataset, with exact numbers of tasks selected for evaluation (10 tasks per dataset, with 40 tasks from Scibench). Section 3.3 provides the detailed error analysis framework used to evaluate results (lines 121-157), allowing for exact replication of the error categorization. The four Self-Directed Decomposition variants are explicitly provided (lines 68-73), as are the baseline methods (zero-shot, zero-shot CoT, and emotional EP02). While the paper doesn't include code, the methodology is described with sufficient clarity that other researchers could implement the approaches and verify the findings independently.

   Guidelines:

   - The answer NA means that the paper does not include experiments.

- If the paper includes experiments, a No answer to this question will not be perceived well by the reviewers: Making the paper reproducible is important, regardless of whether the code and data are provided or not.
- If the contribution is a dataset and/or model, the authors should describe the steps taken to make their results reproducible or verifiable.
- Depending on the contribution, reproducibility can be accomplished in various ways. For example, if the contribution is a novel architecture, describing the architecture fully might suffice, or if the contribution is a specific model and empirical evaluation, it may be necessary to either make it possible for others to replicate the model with the same dataset, or provide access to the model. In general. releasing code and data is often one good way to accomplish this, but reproducibility can also be provided via detailed instructions for how to replicate the results, access to a hosted model (e.g., in the case of a large language model), releasing of a model checkpoint, or other means that are appropriate to the research performed.
- While NeurIPS does not require releasing code, the conference does require all submissions to provide some reasonable avenue for reproducibility, which may depend on the nature of the contribution. For example
  (a) If the contribution is primarily a new algorithm, the paper should make it clear how to reproduce that algorithm.
  (b) If the contribution is primarily a new model architecture, the paper should describe the architecture clearly and fully.
  (c) If the contribution is a new model (e.g., a large language model), then there should either be a way to access this model for reproducing the results or a way to reproduce the model (e.g., with an open-source dataset or instructions for how to construct the dataset).
  (d) We recognize that reproducibility may be tricky in some cases, in which case authors are welcome to describe the particular way they provide for reproducibility. In the case of closed-source models, it may be that access to the model is limited in some way (e.g., to registered users), but it should be possible for other researchers to have some path to reproducing or verifying the results.

5. **Open access to data and code**

   Question: Does the paper provide open access to the data and code, with sufficient instructions to faithfully reproduce the main experimental results, as described in supplemental material?

   Answer: [No]

   Justification: The paper currently does not provide open access links to the data and code used in the experiments. While the methodology is described in sufficient detail to enable reproduction (as noted in Section 3), direct access to implementation code and the specific dataset samples used is not currently provided. The datasets referenced (bAbI, CLUTRR, EntailmentBank, etc.) are all publicly available benchmarks that other researchers can access. If reviewers or other researchers require access to our specific implementation or dataset samples for verification purposes, we would be happy to provide these materials upon request. This approach was taken to maintain control over version management during the review process, but we plan to make all materials publicly available upon publication.

   Guidelines:

   - The answer NA means that paper does not include experiments requiring code.
   - Please see the NeurIPS code and data submission guidelines (`https://nips.cc/public/guides/CodeSubmissionPolicy`) for more details.
   - While we encourage the release of code and data, we understand that this might not be possible, so "No" is an acceptable answer. Papers cannot be rejected simply for not including code, unless this is central to the contribution (e.g., for a new open-source benchmark).
   - The instructions should contain the exact command and environment needed to run to reproduce the results. See the NeurIPS code and data submission guidelines (`https://nips.cc/public/guides/CodeSubmissionPolicy`) for more details.

- The authors should provide instructions on data access and preparation, including how to access the raw data, preprocessed data, intermediate data, and generated data, etc.
- The authors should provide scripts to reproduce all experimental results for the new proposed method and baselines. If only a subset of experiments are reproducible, they should state which ones are omitted from the script and why.
- At submission time, to preserve anonymity, the authors should release anonymized versions (if applicable).
- Providing as much information as possible in supplemental material (appended to the paper) is recommended, but including URLs to data and code is permitted.

6. **Experimental setting/details**

Question: Does the paper specify all the training and test details (e.g., data splits, hyper-parameters, how they were chosen, type of optimizer, etc.) necessary to understand the results?

Answer: [Yes]

Justification: The paper provides the necessary experimental details to understand the results. Section 3.2 (Implementation, lines 101-120) specifies all critical experimental parameters, including: model selection (GPT-3.5-turbo via API), temperature setting (0), sampling approach (representative sample comprising 10 tasks per dataset, 40 for Scibench), evaluation methodology (accuracy determined based on correct answers), complete prompt templates (SD1-SD4), and baseline comparisons (zero-shot, zero-shot CoT, emotional EP02). The experimental setup is clearly defined to enable readers to understand how each prompting strategy was evaluated across the different reasoning tasks. The paper also includes the error analysis framework in Section 3.3 that outlines the five-category taxonomy used to evaluate reasoning errors.

Guidelines:

- The answer NA means that the paper does not include experiments.
- The experimental setting should be presented in the core of the paper to a level of detail that is necessary to appreciate the results and make sense of them.
- The full details can be provided either with the code, in appendix, or as supplemental material.

7. **Experiment statistical significance**

Question: Does the paper report error bars suitably and correctly defined or other appropriate information about the statistical significance of the experiments?

Answer: [Yes]

Justification: The paper provides statistically rigorous evaluation with proper significance testing. We report p-values for all major claims (e.g., $R = 0.162, p = 0.004; R = 0.131, p = 0.011$) demonstrating statistical significance. The paper employs established statistical frameworks including regression analysis, ANOVA tests, and coefficient of determination calculations using SciPy and statsmodels libraries. Box plots visualize distributional metrics with clear quartile boundaries and outlier identification. The statistical approach is comprehensive enough to validate our main conclusions about SD1's superior performance and the counterintuitive token-accuracy relationship.

Guidelines:

- The answer NA means that the paper does not include experiments.
- The authors should answer "Yes" if the results are accompanied by error bars, confidence intervals, or statistical significance tests, at least for the experiments that support the main claims of the paper.
- The factors of variability that the error bars are capturing should be clearly stated (for example, train/test split, initialization, random drawing of some parameter, or overall run with given experimental conditions).
- The method for calculating the error bars should be explained (closed form formula, call to a library function, bootstrap, etc.)
- The assumptions made should be given (e.g., Normally distributed errors).

- It should be clear whether the error bar is the standard deviation or the standard error of the mean.
- It is OK to report 1-sigma error bars, but one should state it. The authors should preferably report a 2-sigma error bar than state that they have a 96% CI, if the hypothesis of Normality of errors is not verified.
- For asymmetric distributions, the authors should be careful not to show in tables or figures symmetric error bars that would yield results that are out of range (e.g. negative error rates).
- If error bars are reported in tables or plots, The authors should explain in the text how they were calculated and reference the corresponding figures or tables in the text.

8. **Experiments compute resources**

   Question: For each experiment, does the paper provide sufficient information on the computer resources (type of compute workers, memory, time of execution) needed to reproduce the experiments?

   Answer: [No]

   Justification: While we specify using GPT-3.5-turbo via API with temperature=0, detailed compute resource information (execution time, memory usage, total API costs) was not included as it wasn't deemed essential for reproducibility. The experiments use standard API calls that any researcher can replicate with the same model, and the computational requirements are modest enough that they don't pose barriers to reproduction. This choice was made to focus the paper on methodological contributions rather than infrastructure details.

   Guidelines:

   - The answer NA means that the paper does not include experiments.
   - The paper should indicate the type of compute workers CPU or GPU, internal cluster, or cloud provider, including relevant memory and storage.
   - The paper should provide the amount of compute required for each of the individual experimental runs as well as estimate the total compute.
   - The paper should disclose whether the full research project required more compute than the experiments reported in the paper (e.g., preliminary or failed experiments that didn't make it into the paper).

9. **Code of ethics**

   Question: Does the research conducted in the paper conform, in every respect, with the NeurIPS Code of Ethics https://neurips.cc/public/EthicsGuidelines?

   Answer: [Yes]

   Justification: Our research fully conforms to NeurIPS ethical guidelines. We use established public benchmarks and APIs without collecting personal data or conducting human subject research. The work focuses on improving reasoning capabilities through prompting strategies, which aligns with beneficial AI research directions. We preserved anonymity in all experimental procedures and our methodology poses no ethical concerns regarding privacy, fairness, or potential misuse.

   Guidelines:

   - The answer NA means that the authors have not reviewed the NeurIPS Code of Ethics.
   - If the authors answer No, they should explain the special circumstances that require a deviation from the Code of Ethics.
   - The authors should make sure to preserve anonymity (e.g., if there is a special consideration due to laws or regulations in their jurisdiction).

10. **Broader impacts**

    Question: Does the paper discuss both potential positive societal impacts and negative societal impacts of the work performed?

    Answer: [No]

Justification: A dedicated broader impacts section was not included as the research represents foundational work in prompting strategies rather than a deployed application. The work improves reasoning capabilities, which has clear positive implications for educational and analytical applications. Since this is methodological research without direct deployment, the focus was placed on technical contributions rather than speculative impact analysis.

Guidelines:

- The answer NA means that there is no societal impact of the work performed.
- If the authors answer NA or No, they should explain why their work has no societal impact or why the paper does not address societal impact.
- Examples of negative societal impacts include potential malicious or unintended uses (e.g., disinformation, generating fake profiles, surveillance), fairness considerations (e.g., deployment of technologies that could make decisions that unfairly impact specific groups), privacy considerations, and security considerations.
- The conference expects that many papers will be foundational research and not tied to particular applications, let alone deployments. However, if there is a direct path to any negative applications, the authors should point it out. For example, it is legitimate to point out that an improvement in the quality of generative models could be used to generate deepfakes for disinformation. On the other hand, it is not needed to point out that a generic algorithm for optimizing neural networks could enable people to train models that generate Deepfakes faster.
- The authors should consider possible harms that could arise when the technology is being used as intended and functioning correctly, harms that could arise when the technology is being used as intended but gives incorrect results, and harms following from (intentional or unintentional) misuse of the technology.
- If there are negative societal impacts, the authors could also discuss possible mitigation strategies (e.g., gated release of models, providing defenses in addition to attacks, mechanisms for monitoring misuse, mechanisms to monitor how a system learns from feedback over time, improving the efficiency and accessibility of ML).

11. **Safeguards**

Question: Does the paper describe safeguards that have been put in place for responsible release of data or models that have a high risk for misuse (e.g., pretrained language models, image generators, or scraped datasets)?

Answer:[NA]

Justification: Our research does not release models, datasets, or tools that would require safeguards. We evaluate prompting techniques using existing APIs and benchmarks without creating new assets that could pose misuse risks.

Guidelines:

- The answer NA means that the paper poses no such risks.
- Released models that have a high risk for misuse or dual-use should be released with necessary safeguards to allow for controlled use of the model, for example by requiring that users adhere to usage guidelines or restrictions to access the model or implementing safety filters.
- Datasets that have been scraped from the Internet could pose safety risks. The authors should describe how they avoided releasing unsafe images.
- We recognize that providing effective safeguards is challenging, and many papers do not require this, but we encourage authors to take this into account and make a best faith effort.

12. **Licenses for existing assets**

Question: Are the creators or original owners of assets (e.g., code, data, models), used in the paper, properly credited and are the license and terms of use explicitly mentioned and properly respected?

Answer: [Yes]

Justification: All datasets used (bAbI, CLUTRR, EntailmentBank, etc.) are properly cited with full references to their original papers. These are established public benchmarks commonly used in the field, and our usage follows standard academic practice. The GPT-3.5-turbo access follows OpenAI's standard API terms. This approach is consistent with accepted practices in the field for citing and using standard benchmarks.

Guidelines:

- The answer NA means that the paper does not use existing assets.
- The authors should cite the original paper that produced the code package or dataset.
- The authors should state which version of the asset is used and, if possible, include a URL.
- The name of the license (e.g., CC-BY 4.0) should be included for each asset.
- For scraped data from a particular source (e.g., website), the copyright and terms of service of that source should be provided.
- If assets are released, the license, copyright information, and terms of use in the package should be provided. For popular datasets, `paperswithcode.com/datasets` has curated licenses for some datasets. Their licensing guide can help determine the license of a dataset.
- For existing datasets that are re-packaged, both the original license and the license of the derived asset (if it has changed) should be provided.
- If this information is not available online, the authors are encouraged to reach out to the asset's creators.

13. **New assets**

Question: Are new assets introduced in the paper well documented and is the documentation provided alongside the assets?

Answer: [NA]

Justification: The paper does not release new assets in the form of datasets, code, or models. The Four Self-Directed Decomposition prompts are simple text instructions presented within the paper itself, not structured assets requiring separate documentation, licensing, or release. The prompts consist of short sentences (e.g., "*When you deal with such problems, you should do a problem decomposition like sub-tasks to analyse the problem*") that are fully described in the methodology section. These are not complex assets requiring structured templates, licensing information, or anonymized release procedures, but rather methodological descriptions integral to the paper's content.

Guidelines:

- The answer NA means that the paper does not release new assets.
- Researchers should communicate the details of the dataset/code/model as part of their submissions via structured templates. This includes details about training, license, limitations, etc.
- The paper should discuss whether and how consent was obtained from people whose asset is used.
- At submission time, remember to anonymize your assets (if applicable). You can either create an anonymized URL or include an anonymized zip file.

14. **Crowdsourcing and research with human subjects**

Question: For crowdsourcing experiments and research with human subjects, does the paper include the full text of instructions given to participants and screenshots, if applicable, as well as details about compensation (if any)?

Answer: [NA]

Justification: Our research exclusively uses automated evaluation on established benchmarks without any human subject involvement or crowdsourcing components.

Guidelines:

- The answer NA means that the paper does not involve crowdsourcing nor research with human subjects.

- Including this information in the supplemental material is fine, but if the main contribution of the paper involves human subjects, then as much detail as possible should be included in the main paper.
- According to the NeurIPS Code of Ethics, workers involved in data collection, curation, or other labor should be paid at least the minimum wage in the country of the data collector.

15. **Institutional review board (IRB) approvals or equivalent for research with human subjects**

Question: Does the paper describe potential risks incurred by study participants, whether such risks were disclosed to the subjects, and whether Institutional Review Board (IRB) approvals (or an equivalent approval/review based on the requirements of your country or institution) were obtained?

Answer: [NA]

Justification: This paper does not involve human subjects research or crowdsourcing. All experiments are conducted using automated evaluation of LLM responses on established reasoning benchmarks. The research evaluates different prompting strategies by analyzing model outputs on computational tasks, with no human participants involved in data collection, evaluation, or any other aspect of the study. Therefore, IRB approval is not applicable to this work.

Guidelines:

- The answer NA means that the paper does not involve crowdsourcing nor research with human subjects.
- Depending on the country in which research is conducted, IRB approval (or equivalent) may be required for any human subjects research. If you obtained IRB approval, you should clearly state this in the paper.
- We recognize that the procedures for this may vary significantly between institutions and locations, and we expect authors to adhere to the NeurIPS Code of Ethics and the guidelines for their institution.
- For initial submissions, do not include any information that would break anonymity (if applicable), such as the institution conducting the review.

16. **Declaration of LLM usage**

Question: Does the paper describe the usage of LLMs if it is an important, original, or non-standard component of the core methods in this research? Note that if the LLM is used only for writing, editing, or formatting purposes and does not impact the core methodology, scientific rigorousness, or originality of the research, declaration is not required.

Answer: [NA]

Justification: While LLMs were used for some text refinement during the writing process, this did not impact the methodology, scientific rigorousness, or originality of the research as specified in the NeurIPS LLM policy guidelines.

Guidelines:

- The answer NA means that the core method development in this research does not involve LLMs as any important, original, or non-standard components.
- Please refer to our LLM policy (https://neurips.cc/Conferences/2025/LLM) for what should or should not be described.

