# OpenReview forum: "Self-Directed Decomposition Empowers Reasoning Potentials in Large Language Models"
_NeurIPS.cc/2025/Conference — Submitted to NeurIPS 2025_

### Official Review · Reviewer_fw5S · 2025-06-01

**Clarity:** 1
**Significance:** 1
**Originality:** 1
**Rating:** 1
**Confidence:** 4

**Summary:**

This paper presents Self-Directed Decomposition (SD), a prompting strategy that empowers large language models (LLMs) to autonomously break down complex reasoning problems into sub-tasks, evaluating four variants (SD1-SD4). Experiments across seven reasoning tasks demonstrate that SD, particularly SD1, enhances performance on deductive, inductive, and mathematical reasoning, while uncovering trade-offs in error patterns and a counterintuitive negative correlation between token usage and accuracy.

**Questions:**

- Could the authors clarify how SD differs conceptually from prior task-decomposition prompting methods to establish novelty, and provide comparative performance data against these stronger baselines?
- Regarding readability, can the authors enlarge Figure 1’s font or provide a zoomable version in supplements, and ensure all acronyms (e.g., SD1) are defined before first use in the main text?
- To address experimental generalizability, will the authors validate SD on open-source models (e.g., Llama 2, Mistral) or larger-scale models (e.g., GPT-4) to demonstrate cross-model effectiveness?
- Given the weak baselines, would adding comparisons with state-of-the-art decomposition methods (e.g., Task Dynamic Decomposition, Graph of Thoughts) strengthen the claim of SD’s efficacy?

**Ethical Concerns:**

["NO or VERY MINOR ethics concerns only"]

**Limitations:**

yes

**Quality:**

2

**Strengths And Weaknesses:**

- The innovativeness is limited. The paper proposes using prompts to enable LLMs to decompose complex tasks into subtasks for reasoning, which has already been extensively explored in this field. In terms of motivation, this paper lacks originality. Regarding the specific methods, the paper only conducts experimental analyses on four different prompt contents without deeper exploration.
- The readability is poor. For example, the font in Figure 1 is too small for readers to read the content. The logical flow of the paper is also weak. For instance, the Abstract mentions "SD1 maintaining optimal balance," but what SD1 is remains undefined in the preceding content. There are multiple such issues throughout the paper, which make it difficult to read and comprehend.
- The experimental setup is unreasonable. The authors only conduct experiments on a closed-source model, GPT-3.5-turbo, which makes it impossible to guarantee that the experimental conclusions can be generalized to other models, thus lacking credibility. The baseline methods used by the authors (zero-shot, zero-shot CoT, or emotional EP02) are relatively weak. Outperforming these methods does not demonstrate that the proposed method is significantly effective.
- Overall, this manuscript is more like a technical report or project assignment for a certain course rather than an article that could be considered for publication by NeurIPS.

---

### Official Review · Reviewer_HDBF · 2025-06-20

**Clarity:** 2
**Significance:** 2
**Originality:** 1
**Rating:** 2
**Confidence:** 4

**Summary:**

This paper prompts LLM to conduct a divide-and-conquer style of reasoning and make analyses across various benchmarks, comparing them with basic chain-of-thought prompting. They also discovered the counter-intuitive conclusion that there is a negative correlations between the token consumption and accuracy.

**Questions:**

* For the "More Computational Resources Don’t Guarantee Better Performance", I think the author might also want to discuss the computational complexity of the problem. If a reasoning problem is computationally hard (such as an NP-hard puzzle), in some cases, it might be hard to compute the optimal answer in a very efficient way (especially in the worst case) without more computation. Increased token consumption may indicate more computation and should lead to improved performance. As such, there should be a trade-off between the final accuracy and accumulated process mistakes. Is it possible to see an improvement in the performance by consuming more tokens, then a decrease after more process mistakes occur?
* The policy produces suboptimal reasoning trajectories to compute and get the final answer, such as simulating a brute-force search process. The SD method (or divide-and-conquer) is general, but might not result in optimal solutions in many cases (such as arithmetic computation, which has linear complexity, or linear dynamic programming in [2]). This would result in more unnecessary steps and in more process mistakes. I am curious about your opinion.

**Ethical Concerns:**

["NO or VERY MINOR ethics concerns only"]

**Limitations:**

Yes

**Paper Formatting Concerns:**

No.

**Quality:**

2

**Strengths And Weaknesses:**

**Strength**
* The idea is simple and general to be applied to various reasoning domains. The divide-and-conquer is a general problem-solving method, algorithmically and philosophically.
* The experimental analysis is interesting. It has very detailed quantitative analysis about the result, showing statistical significance of the conclusion.

**Weaknesses**
* The paper is not technically strong. The core idea is based on prompt engineering, which has limited technical contribution. The experiment only uses GPT-3.5, which is somewhat outdated, and the landscape is changing rapidly. The conclusion may be affected if it were applied to different, state-of-the-art language models.
* The novelty is limited and outdated. Some papers have conducted very similar studies, such as [1], and their evaluation is more comprehensive with more baselines. The author may need to articulate their contribution with respect to the counterpart of the paper.
* Many of the conclusions overlap with published papers. For example, some of the conclusions, such as More Computational Resources Don’t Guarantee Better Performance are inspiring, but have already been studied by [2]. The author might need to have a more comprehensive review of related work.
* The writing could be improved. There are so many abbreviations that they might compromise the readability. The author might want to use it sparingly.

[1] https://arxiv.org/pdf/2402.05359

[2] https://arxiv.org/pdf/2305.18654

---

### Official Review · Reviewer_nGtX · 2025-07-03

**Clarity:** 2
**Significance:** 1
**Originality:** 1
**Rating:** 2
**Confidence:** 5

**Summary:**

This paper proposes Self-Directed Decomposition (SD), a prompt-based method that can decompose reasoning problems into sub-tasks. The authors propose four prompt variants (SD1-SD4) with decreasing levels of structural guidance, ranging from explicit decomposition instructions to minimal prompts. The methodology is evaluated across seven reasoning domains (deductive, inductive, mathematical, scientific, commonsense, abductive, and causal) using GPT-3.5-turbo. Comparing to zero-shot inference or zero-shot chain-of-thought approaches, SD1 achieves more than 15% improvement.

**Questions:**

- Why are the variances of SD methods' performances high? Especially SD2 and SD3.
- The results are solely in figures, making it hard to read and harder to compare with other papers and methods. Please use tables to present some of the results to make it clearer.

**Ethical Concerns:**

["NO or VERY MINOR ethics concerns only"]

**Limitations:**

As mentioned in the weaknesses section, please address the limitations in evaluation (small sample size, only one model) and experimentation (limited baseline model comparisons).

**Quality:**

2

**Strengths And Weaknesses:**

## Strengths
- This paper applies the existing five-category error taxonomy to understand how SD affects different reasoning types and token lengths.
- R^2 and p-values are used for most results, suggesting the effort in result analysis.

## Weaknesses

- The evaluation is extremely limited. The proposed method, SD, is purely prompt-based. However, the paper only evaluates the prompts on a single model (GPT-3.5 Turbo). Furthermore, only 10 tasks per dataset were selected (except Scibench).  With this small scale evaluation and no cross-model validation, it is not that convincing that SD helps.
- Furthermore, the paper fails to compete against other popular prompt-based methods like self-consistency or tree-of-thoughts. Although the paper claims that these methods operates on predefined exemplars or researcher-designed algorithms, I still think it is valuable to compare SD against. However, SD also consists of 4 prompts that are also researcher-designed.
- Zhou et al. (2024), which proposes self-discover, an algorithm that enable LLMs to self-discover the task-intrinsic reasoning structures, seems to have a concept overlap with this paper. Zhou et al., (2024) demonstrated the method’s effectiveness on multiple model families, and provides a more statistically significant analysis of the method.

---

### Official Review · Reviewer_yVmF · 2025-07-03

**Clarity:** 2
**Significance:** 3
**Originality:** 1
**Rating:** 2
**Confidence:** 4

**Summary:**

This paper investigates 4 different prompting strategies that decompose the task into sub-goals and attempts to solve them. The authors introduce four zero-shot Self-Directed Decomposition (SD) prompts (SD1-SD4) designed to make GPT-3.5 break problems into sub-tasks without examples. They evaluate these against vanilla zero-shot, zero-shot CoT, and an emotional prompt across seven reasoning benchmarks and find SD1 yields the highest accuracy. A cross-dataset token-accuracy analysis and detailed error breakdown show that effective decomposition must balance reducing selection mistakes with avoiding overly long reasoning chains.

**Questions:**

See weaknesses above.

**Ethical Concerns:**

["NO or VERY MINOR ethics concerns only"]

**Limitations:**

yes

**Paper Formatting Concerns:**

* The figures break when enlarging them.

* The citation should be in the [] bracket, not () parenthesis.

**Quality:**

2

**Strengths And Weaknesses:**

* (Strength) In Section 3.3, the authors adopts a five-category taxonomy (Wrong Selection, Hallucination, No Reasoning, Perspective Mistake, Process Mistake) to dissect how each SD variant reshapes the model’s reasoning behavior. Such fine-grained categorization makes it easier to diagnose in which aspect the model is failing.

* (Weakness 1) While mentioned in the introduction, the authors never pit their SD directly against established zero-shot decomposition methods like Plan-and-Solve[1], Self-Ask[2], or Least-to-Most[3]. Specifically, in the introduction, the paper surveys a range of CoT-style techniques- Least-to-Most prompting, self-consistency, Logical CoT, Tree of Thoughts, Graph of Thoughts, etc., but then evaluates SD only against three baselines: vanilla zero-shot, zero-shot CoT, and one emotional prompt (EP02). Plan-and-Solve (which is cited in the bibliography) and Self-Ask never appear as baselines in any experiment, nor does Least-to-Most prompting get evaluated head-to-head. That leaves a gap: methods that also do zero-shot task decomposition without demonstrations aren’t directly compared, even though they share the same motivation.

* (Weakness 2) The analysis from Section 4.2 is not reasonable: it pools all tasks when plotting "tokens vs. accuracy", masking task-specific effects. Section 4.2 ("Token-Accuracy Paradox") shows a single regression of accuracy against token count across all seven datasets, reporting a weak but significant negative trend. But because GPT-3.5’s baseline performance differs dramatically by task (for instance, SciBench is much harder than Math or CommonsenseQA), combining them can create spurious downward slopes. A fairer test would plot token-accuracy for each dataset separately - e.g. SD vs. zero-shot vs. CoT on SciBench alone - so we see whether longer outputs really hurt within each task rather than across mismatched tasks.

[1] Plan-and-Solve Prompting: Improving Zero-Shot Chain-of-Thought Reasoning by Large Language Models (https://arxiv.org/abs/2305.04091)
[2] Measuring and Narrowing the Compositionality Gap in Language Models (https://arxiv.org/abs/2210.03350)
[3] Least-to-Most Prompting Enables Complex Reasoning in Large Language Models (https://arxiv.org/abs/2205.10625)

---

### Decision · Program_Chairs · 2025-09-17

**Decision:**

Reject

**Comment:**

This paper proposes Self-Directed Decomposition (SD), a prompt-based method that can decompose reasoning problems into sub-tasks. The authors propose four prompt variants (SD1-SD4) with decreasing levels of structural guidance, ranging from explicit decomposition instructions to minimal prompts. The methodology is evaluated across seven reasoning domains by comparing with zero-shot and zero-shot CoT methods.

Strengths:

The idea is simple and general to be applied to various reasoning domains. The experimental analysis is interesting, such as applies the existing five-category error taxonomy to understand how SD affects different reasoning types and token lengths.

Weakness:

All reviewers concern about the evaluation setting such as used baselines, there are tons of decompositon-based prompting method such as Self-Ask, Plan-and-Solve, and Least-to-Most. None of them is compared, and some reviewers also raise the issue of analysis (i.e., fair plot), generalization capability (i.e., more backbone models), writing and readability.